# Mixing Enhancement of a Passive Micromixer with Submerged Structures

**DOI:** 10.3390/mi13071050

**Published:** 2022-06-30

**Authors:** Makhsuda Juraeva, Dong Jin Kang

**Affiliations:** School of Mechanical Engineering, Yeungnam University, Gyoungsan 712-749, Korea; mjuraeva@ynu.ac.kr

**Keywords:** degree of mixing (DOM), submerged structures, Norman window, rectangular baffles, circular passage, vortex burst

## Abstract

A passive micromixer combined with two different mixing units was designed by submerging planar structures, and its mixing performance was simulated over a wider range of the Reynolds numbers from 0.1 to 80. The two submerged structures are a Norman window and rectangular baffles. The mixing performance was evaluated in terms of the degree of mixing (DOM) at the outlet and the required pressure load between inlet and outlet. The amount of submergence was varied from 30 μm to 70 μm, corresponding to 25% to 58% of the micromixer depth. The enhancement of mixing performance is noticeable over a wide range of the Reynolds numbers. When the Reynolds number is 10, the DOM is improved by 182% from that of no submergence case, and the required pressure load is reduced by 44%. The amount of submergence is shown to be optimized in terms of the DOM, and the optimum value is about 40 μm. This corresponds to a third of the micromixer depth. The effects of the submerged structure are most significant in the mixing regime of convection dominance from Re = 5 to 80. In a circular passage along the Norman window, one of the two Dean vortices burst into the submerged space, promoting mixing in the cross-flow direction. The submerged baffles in the semi-circular mixing units generate a vortex behind the baffles that contributes to the mixing enhancement as well as reducing the required pressure load.

## 1. Introduction

In recent years, microfluidics technology has progressed rapidly and has been integrated in many engineering and biomedical applications [1,2]. Micro-mixing is one of the practical technologies used in various microfluidic applications, such as micro-reactors and micro total analysis systems. As the mixing is governed by molecular diffusion, slow fluid velocity and microscale geometry, it corresponds to a very low Reynolds number regime. A micromixer based on molecular diffusion only is not satisfactory for rapid and homogeneous mixing. Various strategies have been proposed to enhance the efficiency of microfluidic mixing, and mixing enhancement is still a crucial design goal [2].

Micromixers are traditionally classified as either active or passive. One of the technologies for increasing mixing efficiency is utilizing an external energy source to create flow disturbances. Various energy sources have been attempted to enhance mixing: acoustic [3], magnetic [4], electric [5], thermal [6], and pressure [7]. As each active micromixer employs an external energy source, the corresponding structure of an active micromixer becomes more complicated than that of a passive micromixer. On the contrary, passive micromixers use a geometric structure to generate a chaotic flow field. As passive micromixers have no moving parts, they are much simpler to integrate into a microfluidic system. Diverse design concepts have been attempted to generate a chaotic flow field. Some examples include staggered herringbone [8], repeated surface groove and baffles [9,10], channel wall twisting [11], block in the junction [12], split-and-recombine (SAR) [13], Tesla structure [14] and stacking of mixing units in the cross flow direction [15].

There are several approaches to enhance the degree of mixing (DOM): complex three dimensional structures, modification of planar geometry, and manipulation of flow conditions. Using an unequal magnitude of inlet velocities is an example to control flow conditions [16,17]. For example, Karvelas et al. [16] showed that a velocity ratio increase leads to an enhancement of mixing between the heavy metals and nanoparticles by the action of shear stress between them. Another example is to use a pulsatile inlet flow. For example, McDonough et al. [17] showed that the micromixing time decreased with an increased velocity ratio of oscillatory velocity to net velocity in all baffle designs. However, this kind of approach usually requires a complicated design of micromixer, compared with a typical planar micromixer. This paper focused on a simpler approach to enhance the DOM, based on a geometric modification.

In general, a complex three-dimensional (3D) micromixer may show a better mixing performance than a two-dimensional (2D) micromixer of similar size [18]. However, 2D planar micromixers are simpler to fabricate, and some planar designs show effective 3D flow characteristics: baffles, channel wall twisting, wall protrusion, spiral channel, and split-and-recombine (SAR). For example, Kang [10] showed that a cyclic attachment of baffles to the micromixer walls generates a rotational flow in the cross section of a micromixer. According to Tsai et al. [19], radial baffles implemented in a circular channel induce vortices in multiple directions. Santana et al. [20] used triangular baffles to promote split-and-recombine of streams and vortex generation. Sotowa et al. [21] used indentations and baffles to generate a secondary flow in deep microchannel reactors. Chung et al. [22] proposed short planar baffles with a gap in the cross flow direction and obtained a strong vortex flow inside the mixing units. Tseng et al. [23] used wall protrusions to obtain a strong vortex for the Reynolds numbers Re ≥ 10. Tripathi et al. [24] used a spiral channel to generate the Dean vortex and showed that the mixing performance is quite dependent on the micromixer layout. Li et al. [25] combined a planar asymmetric SAR with dislocated sub-channels, and showed that multiple vortices are formed in the dislocated sub-channels. Makhsuda et al. [26,27] showed that a cross channel SAR generates a vortex in the transverse direction as well as the Dean vortex.

Most passive micromixers are based on the Dean vortex or vortices by obstacles such as baffles. However, these flow characteristics are usually dominant only in a limited range of the Reynolds number and/or require higher pressure load in the convection dominant regime of mixing. For example, Raza et al. [18] showed that the Dean vortex is one of most practical flow characteristics for the Reynolds numbers Re ≥ 40. On the contrary, the mixing in the diffusion dominant regime (Re ≤ 1) can be enhanced by increasing the residential time of the working fluid. Therefore, a micromixer geometry should be designed for the working fluid to take a longer flow path. In the transition regime of mixing, a secondary flow becomes significant and may lead to mixing enhancement. However, the use of additional obstacles to generate secondary flow increases the required pressure load.

Many micromixers used in the biological and chemical applications operate usually in the range of millisecond mixing time, and the corresponding Reynolds number is less than about 100 [28,29,30]. In this range, the mixing mechanism is usually divided into three regimes: molecular dominance, transition, and convection dominance [18,26]. Accordingly, the present numerical study was carried out to cover all of the three mixing regimes for the Reynolds numbers from 0.1 to 80. The corresponding volume flow rate ranged from 1.3 μL/min to 1012.8 μL/min.

In this respect, a new design concept to generate a more complicated flow pattern without any additional pressure load is proposed: the submergence of planar structures. The submergence of planar structures has several advantages. First, it does not require any additional complexity of micromixer geometry. This means that a part of micromixer geometry is slightly shorter than the depth of the micromixer. Second, the submergence reduces the blockage effect on the flow, as a part of the planar geometry of the micromixer is submerged; the gap between the planar structure and the micromixer wall is fully open to the flow. Therefore, the submergence of planar structures requires less pressure load compared with that of no submergence planar design. The gap between the planar structure and the channel wall is expected to play a role to generate multiple vortices as well as a secondary flow in the cross section. To verify this design concept, two different types of mixing units are used: Norman window and rectangular baffles. When a Norman window is submerged, it creates a circular passage with its top open; it is formed along the circular part of the Norman window. Therefore, the submergence height of a Norman window may control the intensity of the Dean vortex, especially in the convection dominance regime of mixing. When two rectangular baffles are placed in a semi-circular mixing cell, a vortex is generated in the plane normal to that of the Dean vortex [26]. In addition, the submergence of rectangular baffles may cause another vortex in the third plane normal to the previous two planes. Therefore, a combination of these two mixing units with a submergence is expected to generate complex 3D flow patterns in the mixing units. The gap between the planar structure and the micromixer wall plays a key role and varies within the range of 30 μm to 70 μm. It corresponds to one-quarter to two-thirds of the micromixer depth.

The mixing performance of the present micromixer was analyzed numerically in terms of the degree of mixing (DOM) and the required pressure load. A numerical approach has several advantages. Visualization of the mixing process and the associated flow patterns such as streamlines and vortex formation are easy to obtain. Accordingly, a numerical approach is widely used in studying the mixing performance of a micromixer. For example, Kang [10] used the commercial software ANSYS^®^ Fluent 2020 R2 [31] to simulate the mixing process in a passive micromixer with baffles and evaluated quantitatively the mixing performance. Rhoades et al. [32] used the commercial software COMSOL Multiphysics 5.1 (COMSOL, Inc., Burlington, MA, USA) to study the mixing performance of a grooved serpentine micro-channel. Volpe et al. [33] used the lattice Boltzmann method (LBM) to study the flow dynamics of a continuous size-based sorter microfluidic device.

In this paper, the present micromixer was simulated by using the commercial software ANSYS^®^ Fluent 2020 R2, Canonsburg, PA, USA [31]. The mixing performance was estimated in terms of the degree of mixing (DOM) and corresponding mixing energy cost (MEC) and was compared with those of uncombined micromixers.

## 2. A Passive Micromixer with Submerged Structures

Figure 1 shows the two mixing units to be submerged: circular passage and rectangular baffles in a semi-circle mixing unit. The Norman window shown in Figure 1a has a thickness *W_s_* in the transverse direction, and it results in a circular passage submerged along the micromixer wall. The rectangular baffles shown in Figure 1b allow the fluid flow over the short rectangular baffles and generate another flow passage at the end of baffles. The height of the submerged structure, $W_{s}$, is optimized in terms of the DOM at the outlet.

Figure 2 shows a schematic diagram of the present micromixer. The cross-section of the inlet and outlet branches is a rectangle 300 μm wide and 120 μm deep. Both inlets 1 and 2 are 1250 μm long, while the outlet branch is 1000 μm long. Figure 2a is a three-dimensional view of the present micromixer, and Figure 2b shows the corresponding frontal view. Two inlets are facing opposite to each other, and mixing is expected to take place mainly along the four mixing units. One mixing unit consists of two blocks: One block is made with the Norman window, and the other block has two rectangular baffles inside.

## 3. Governing Equations and Computational Procedure

The fluid was assumed Newtonian and incompressible, and the governing equations are the continuity and Navier–Stokes equations, as follows:(1)$$\left(\vec{u}\cdot \nabla \right)\vec{u}=-\frac{1}{\rho }\nabla p+\nu \nabla^{2}\vec{u},$$
(2)$$\nabla \cdot \vec{u}=0,$$
where $\vec{u}$, *p*, and *ν* are the velocity vector, pressure, and kinematic viscosity, respectively. The evolution of mixing was simulated by solving an advection-diffusion equation:(3)$$\left(\vec{u}\cdot \nabla \right)\varphi =D\nabla^{2}\varphi ,$$
where *D* and *φ* are the mass diffusivity and mass fraction of fluid A, respectively.

The governing Equations (1)–(3) were solved using the commercial software, ANSYS^®^ FLUENT 2020 R2, Canonsburg, PA, USA [31], which is based on the finite volume method. The convective terms in Equations (1) and (3) were discretized using the QUICK scheme (quadratic upstream interpolation for convective kinematics), and its theoretical accuracy is third order. The velocity distribution at the two inlets was assumed uniform, while the outflow condition was specified at the outlet. All other walls were treated as a no-slip boundary. The mass fraction of fluid A was specified *φ* = 1 at inlet 1 and *φ* = 0 at inlet 2.

The mixing performance of a combined micromixer was evaluated using the degree of mixing (*DOM*) and mixing energy cost (*MEC*). The *DOM* is defined in the following form:(4)$$DOM=1-\frac{1}{\xi }\sqrt{\overset{n}{\underset{i=1}{\sum }}\frac{{\left(\varphi_{i}-\xi \right)}^{2}}{n}},$$
where *φ_i_* and *n* are the mass fraction of fluid A in the *i*th cell and the total number of cells, respectively; *ξ* = 0.5, which means equal mixing of the two fluids. The *MEC* was used to evaluate the effectiveness of a micromixer and was defined by combining the pressure load and *DOM* in the following form [34,35]:(5)MEC=∆pρumean2DOM×100,
where $u_{mean}$ is the average velocity at the outlet, and ∆p is the pressure load between the inlet and outlet.

The aqueous fluid flows into the two inlets were assumed to be the same. The fluid properties were assumed to be the same as the physical properties of the water. The density, diffusion constant, and viscosity of the fluid were *ρ* = 997 kg/m^3^, *D* = 1.0 × 10^−10^ m^2^s^−1^, and *ν =* 0.89 × 10^−6^ m^2^s^−^*^1^*, respectively. Therefore, the Schmidt (Sc) number was approximately 10^4^ (the ratio of the kinetic viscosity and the mass diffusivity of the fluid). The Reynolds number was defined as $Re=\frac{\rho U_{mean}d_{h}}{\mu }$, where $\rho ,\;\;U_{mean},\;\;d_{h},\;\mathrm{and}\;\mu$ denote the density, the mean velocity at the outlet, the hydraulic diameter of the outlet channel, and the dynamic viscosity of the fluid, respectively.

## 4. Validation of the Numerical Study

Numerical diffusion is known to deteriorate the accuracy of simulated results for high Sc number simulations in general [36,37,38,39]. Several approaches have been proposed to minimize the numerical diffusion problem in the literature. One example is a particle-based simulation, such as the Monte Carlo method [36], and another approach is to reduce cell Peclet number for grid-based methods. Here, the cell Peclet number is $Pe=\frac{U_{cell}l_{cell}}{D}$ where $U_{cell}$ and $l_{cell}$ are the local flow velocity and cell size, respectively. However, these approaches are computationally too expensive to adopt in a study like this paper. A more practical approach is usually adopted to obtain numerical solutions with a reasonable degree of accuracy. That is to carry a detailed study of grid independence, including the grid convergence index (GCI) test [40,41]. In this paper, a similar procedure was followed.

To validate the present numerical approach, a passive micromixer tested by Tsai et al. [19] was simulated. Figure 3 shows a schematic diagram of the micromixer. The cross section of the two inlets is a rectangle of width 45 μm by depth 130 μm. The micromixer has four baffles of width 40 μm by height 97.5 μm along the circular passage. The fluid was assumed to have the properties of 997 kg/m^3^ density, 0.00097 Ns/m^2^ viscosity, and 3.6 × 10^−10^ m^2^/s diffusion coefficient, as reported by Tsai et al. [19]. A set of numerical simulations was carried out for three different Reynolds numbers: Re *=* 1, 9, and 81. The simulation results were compared with the corresponding experimental data.

The computational domain was meshed with hexahedral cells that were fully structured. Before detailed simulations, a preliminary study was carried out to check the grid independence of numerical solutions for the Reynolds number of 9. Figure 4a shows how the numerical solution depends on the number of cells for Re = 9. When the number of computational cells is larger than about 1 million, the numerical solution shows a reasonable accuracy. Here, *DOM_T_* stands for the degree of mixing defined by Tsai et al. [19] in the following way:(6)$$DOM_{T}=1-\frac{\sigma_{D}}{\sigma_{D,o}},$$
and
(7)$$\sigma =\sqrt{\frac{1}{n}\overset{n}{\underset{i=1}{\sum }}{\left(\phi_{i}-\phi_{ave}\right)}^{2}},$$
where $\sigma_{D}$ is the standard deviation of ϕ on a cross section normal to the flow, $\sigma_{D,o}$ is the standard deviation at the inlet, and $\phi_{ave}\;$ is the average value of ϕ at a sampled cross section.

Figure 4b compares the numerical results quantitatively with the corresponding experimental data by Tsai et al. [19] for Reynolds numbers from 1 to 81. The numerical solution and experimental data shows a similar behavior with the Reynolds number even though there is some discrepancy between them. The discrepancy between the experimental data and numerical solution is less than 4% and becomes smaller as the Reynolds number increases; the Peclet number increases at the same time. The discrepancy is attributed to several factors, such as the numerical diffusion, experimental uncertainty, etc.

Evolution of mixing along the micromixer was also validated by comparing the numerical results with the experimental results obtained by Tsai et al. [19]. Figure 5 compares the numerical result of the concentration distribution on the horizontal mid-plane with the corresponding experimental confocal images for Re *=* 1, 9, and 81. There is acceptable agreement between them for the all Reynolds numbers.

Prior to the present numerical study, an additional set of preliminary simulations was carried out to determine an appropriate mesh size for simulations of the present micromixer. For the present micromixer with four mixing units, the edges of cells were varied from 3 μm to 6 μm. The corresponding number of mesh varied from 1.3 × 10^6^ to 10.2 × 10^6^. The simulation was carried out for Re = 10, and the deviation of 4 μm solution from that of 3 μm was about 0.4%. Therefore, 4 μm was small enough to obtain grid-independent solutions.

Using the numerical solutions, the grid convergence index (GCI) was also calculated to quantify the uncertainty of the grid convergence [40,41]. According to the Richardson extrapolation methodology, the GCI is calculated as follows:(8)$$GCI=F_{s}\frac{\left|\varepsilon \right|}{r^{p}-1},$$
(9)$$\varepsilon =\frac{f_{coarse}-f_{fine}}{f_{fine}},$$
where *F_s_, r,* and *p* are the safety factor of the method, grid refinement ratio, and the order of accuracy of the numerical method, respectively. *f_coarse_* and *f_fine_* are the numerical results obtained with a coarse grid and a fine grid, respectively. *F_s_* is specified as 1.25, as suggested by Roache [40]. For the edge size of 3 μm, 4 μm, and 5 μm, the corresponding number of cells is 10.2 × 10^6^, 4.4 × 10^6^, and 2.2 × 10^6^, respectively. As a result, the GCI of the computed DOM with 4 μm is less than 1%. Therefore, the edge size of *4 μm* was used to obtain numerical solutions with a reasonable accuracy.

## 5. Results and Discussion

A passive micromixer with submerged structures was proposed and simulated to estimate its mixing performance for Reynolds numbers ranging from 0.1 to 80. The velocity at the two inlets was uniform in the range from 0.293 mm/s to 234 mm/s, and the corresponding volume flow rates ranged from 1.3 μL/min to 1012.8 μL/min. Simulations were carried out for four mixing units, and the results were compared with the corresponding solution without any submergence. The degree of mixing was evaluated at the outlet, and the corresponding mixing energy cost was also assessed.

Figure 6 compares the simulated mixing performance of submerged structures in terms of the DOM and the required pressure load with those of the no-submergence case. Here, the case of no submergence means the planar micromixer with a height 120 μm is equal to the depth of the present micromixer. When the height of submerged structures is 70 μm, the corresponding submergence is 50 μm. The results show that the required pressure load with submerged structures is reduced more than 42% from that of no-submergence cases. In addition, the DOM is also noticeably improved over a wide range of Reynolds numbers, as shown in Figure 6a. The maximum enhancement of DOM is 182%, and the corresponding reduction in the pressure load is 44% for the Reynolds number of Re = 10. The effects of submergence on the DOM are clearly observed over a wide range of the Reynolds number, from Re = 5 to 80. The disturbance caused by the submerged structures apparently leads to the mixing enhancement in the mixing regime of convection dominance in terms of both the DOM and pressure load. As expected, the functional relationship between the pressure load and the Reynolds number remains the same. However, there is a noticeable reduction of the pressure load throughout the whole Reynolds number. It suggest that the same value of DOM, even in the molecular diffusion and transition regime of mixing, is achieved with a smaller energy power, as can be seen in Figure 6b. Figure 6c confirms that the submergence of planar structures is more cost-effective than the case of no submergence over a wider range of the Reynolds number.

Figure 7 shows how the amount of submergence affects the mixing performance in terms of the effects of the height of submerged structure *W_s_* on the DOM. The broken horizontal lines in the figure indicate the DOM obtained without any submergence. The DOM shows a great dependence on the height of submerged structures, and it is more significant in the mixing regime of convection dominance; approximately  Re≳5. Particularly noteworthy is the fact that the optimum of submergence exists for the Reynolds numbers  Re≤80. It is almost constant regardless of the Reynolds number and is about one-third of the channel depth. It corresponds to about 80 μm as the height of submerged structure.

To study how the submergence improves the mixing performance in the convection dominance regime of mixing, the flow characteristics in the circular passage are compared on the two cross sections defined in Figure 8. For the case of no submergence, the Dean vortex, a pair of counter rotating vortices, is clearly observed on the planes AA’ and BB’. On the contrary, the left vortex in the circular passage of Figure 8b bursts into the submerged space for the submergence case with *W_s_ =* 60 μm. This flow pattern suggests that additional mixing appears in the transverse direction. As a result, the interface between the two fluids A and B elongates in the transverse direction. This explanation is confirmed in Figure 9, which shows a mixing evolution along the micromixer. For the case of no submergence, fluid A (red in the figure) is enveloped by fluid B (blue in the figure) in the circular channel. This mixing characteristic is due to the Dean vortex developed in the circular passage. On the contrary, an elongated interface between the two fluids is observed for the submerged case, and it is due to the vortex bursting as depicted in Figure 8. This flow characteristic leads to an increase of DOM in the Norman window zone. Figure 10 compares the increment of DOM in each circular channel; refer to the channel numbers depicted in Figure 1a. A significant difference between them is observed, and it is a result of the vortex bursting due to the submergence of the Norman window. For example, the increment of DOM in the first mixing unit for the submerged case is 124% larger than that for the case of no submergence.

A similar improvement of DOM is observed in the semi-circular mixing unit with submerged rectangular baffles. Figure 11 shows projected streamlines and concentration distribution on the cross section CC’; refer to the cross section CC’ in Figure 8. Figure 11a shows that the submergence of baffles generates a vortex behind a baffle in the y-direction, and it causes additional mixing in the z–x plane. This vortex flow promotes the mixing around the second baffle as can be seen in Figure 11c. Comparing Figure 11c with Figure 11b, a strong mixing in the z-direction is observed as a thicker and wider interface between the two fluids before and after the baffles. Figure 12 compares the increment of DOM in each mixing unit of submerged rectangular baffles and shows a noticeable effect of baffle submergence on the DOM. For example, the DOM increment in the first mixing unit for the submerged case is 94% larger than that for no submergence case. Figure 13 compares the concentration distribution in the first semi-circular mixing unit on the plane of half-depth. The flow for the submerged case appears as a wider and wavier interface between the two fluids as it passes over the submerged baffles. This flow pattern promotes the mixing in the transverse direction, and it is confirmed quantitatively as can be seen in Figure 12.

Figure 14 shows the concentration distribution on the three planes: *z =* 30, 60, and 90 μm for the Reynolds number Re = 0.1. The interface between the two fluids becomes wider as they pass through the Norman window at *z* = 60 and 90 μm. This means that the mixing in the transverse direction is active even in the submerged space. This mixing characteristic suggests that the Norman window plays a role as a forward step and spreads the flow in the transverse direction.

## 6. Conclusions

This paper examines how the submergence of a passive micromixer structure enhances the mixing performance of a planar micromixer. The submergence is made by lowering some part of a planar micromixer geometry. In this paper, a passive micromixer is designed by using two different mixing units: one has a circular channel, and the other has rectangular baffles. The present micromixer has eight mixing units, and the submergence is made either by inserting a Norman window in odd-number mixing units or lowering rectangular baffles in even-number mixing units.

The mixing performance was simulated using the commercial software FLUENT 19.2 MEC over a wide range of the Reynolds numbers: Re = 0.1 ~ 80. This range includes three mixing regimes where the dominant mixing mechanism varies from molecular diffusion to convection; both mechanisms are equally significant in the transition regime. The amount of submergence was varied from 30 μm to 70 μm. This corresponds to 25% to 58% of the micromixer depth.

The advantage of submerged structures is observed in terms of the DOM increment and the required pressure load reduction. The enhancement of mixing performance is noticeable over a wide range of the Reynolds numbers Re = 0.1 ~ 80. For example, the DOM is improved 182% from that of the no-submergence case, and the pressure load is reduced 44% for the Reynolds number Re = 10. The amount of submergence is also found to be optimized in terms of the DOM, and the optimum value is about 40 μm, which corresponds to a quarter of the micromixer depth. In the mixing regime of molecular dominance and transition, a noticeable reduction in the required pressure load is achieved with the same value of DOM, compared with that of the no-submergence case. On the contrary, the mixing performance is enhanced in terms of the DOM and the required pressure load in the mixing regime of convection dominance.

The effects of the submerged structure are most significant for the Reynolds numbers Re = 5 to 80. In this range of Reynolds number, the convection effects are crucial, and the submergence of the micromixer structure changes the flow and mixing characteristics. In a circular passage along the Norman window, one of the two Dean vortices burst into the submerged space in a micromixer with submerged structures, and it caused the interface between the two fluids to elongate and widen. This flow characteristic contributes to the mixing enhancement. The submerged baffles in the semi-circular mixing units generate a vortex behind the baffles. It also contributes to the mixing enhancement as well as reducing the required pressure load.

The submergence of a planar micromixer was shown to improve the mixing performance over a wide range of the Reynolds numbers. The improvement of mixing performance is achieved in terms of the DOM enhancement as well as the reduction of the corresponding pressure load. As the submergence is realized by lowering simply some part of micromixer, the concept is applicable for other planar micromixers.

## Figures and Tables

**Figure 1 micromachines-13-01050-f001:**
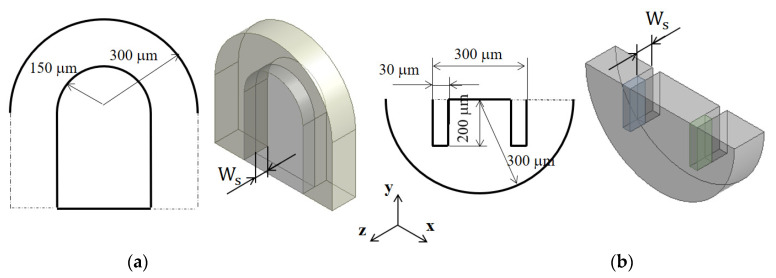
Schematic diagram of two submerged structures (non-proportional): (**a**) Norman window and (**b**) Rectangular baffles.

**Figure 2 micromachines-13-01050-f002:**
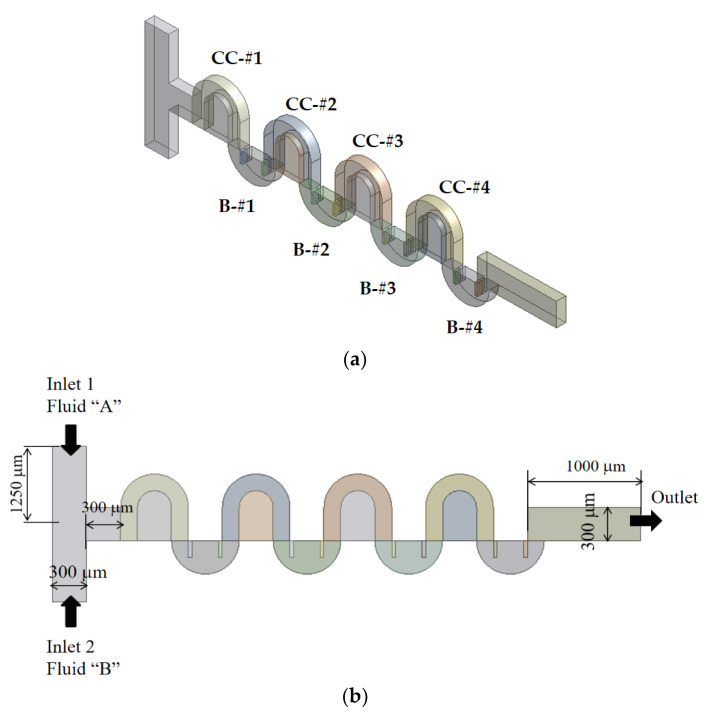
Schematic diagram of present micromixer (non-proportional): (**a**) 3D view and (**b**) Frontal view.

**Figure 3 micromachines-13-01050-f003:**
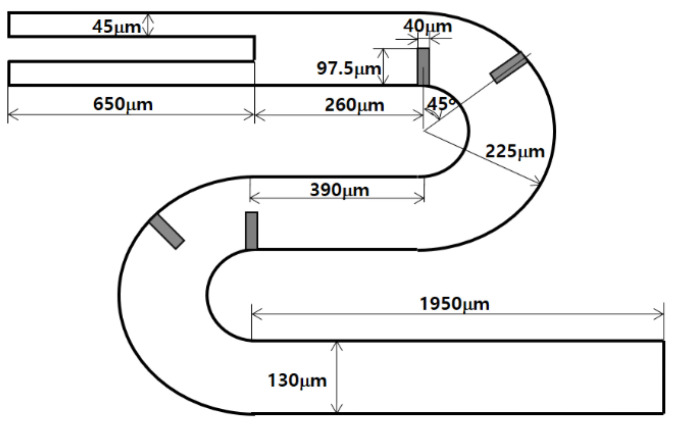
Diagram of the micromixer tested by Tsai et al. [19].

**Figure 4 micromachines-13-01050-f004:**
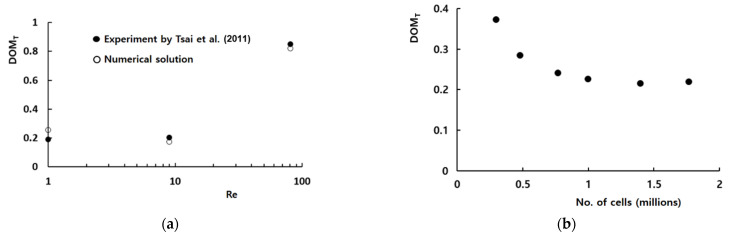
Validation of the numerical solution: (**a**) DOM vs. Re and (**b**) Grid dependence of the numerical solution [19].

**Figure 5 micromachines-13-01050-f005:**
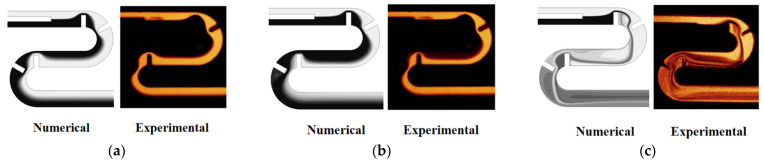
Comparison of numerical concentration distribution with experimental images of Tsai et al. [19]: (**a**) Re = 1, (**b**) Re = 9, and (**c**) Re = 81. Reprinted with permission from [19].

**Figure 6 micromachines-13-01050-f006:**
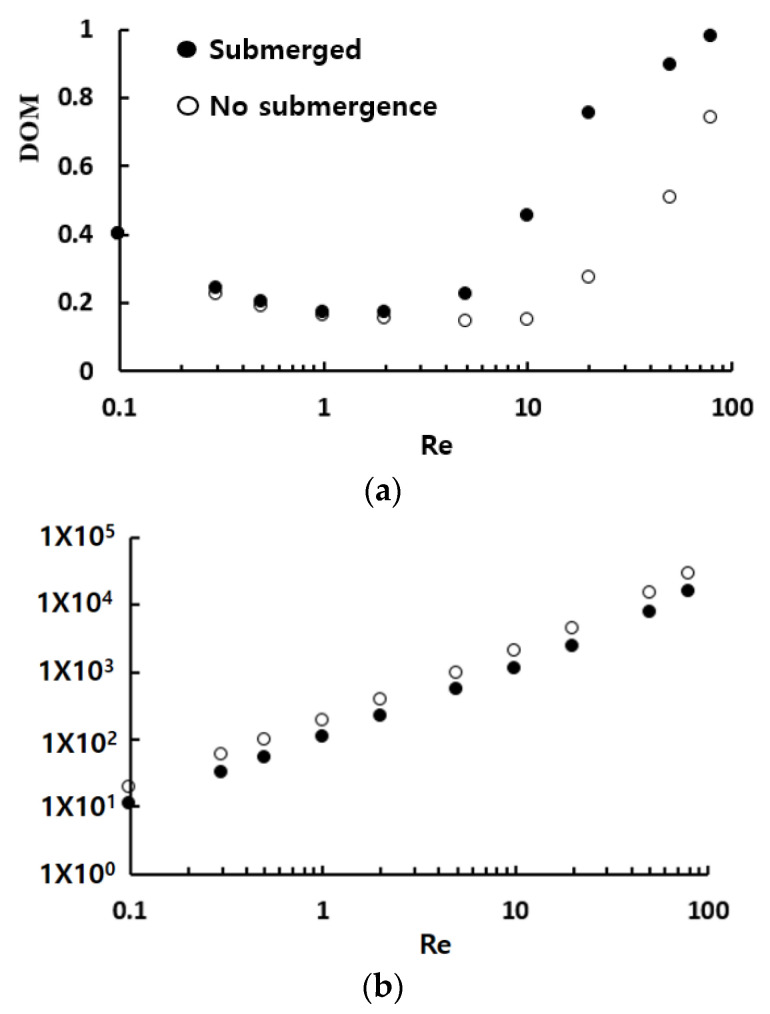
Mixing enhancement of submerged structures: (**a**) DOM vs. Re, (**b**) ΔP vs. Re, and (**c**) MEC vs. Re.

**Figure 7 micromachines-13-01050-f007:**
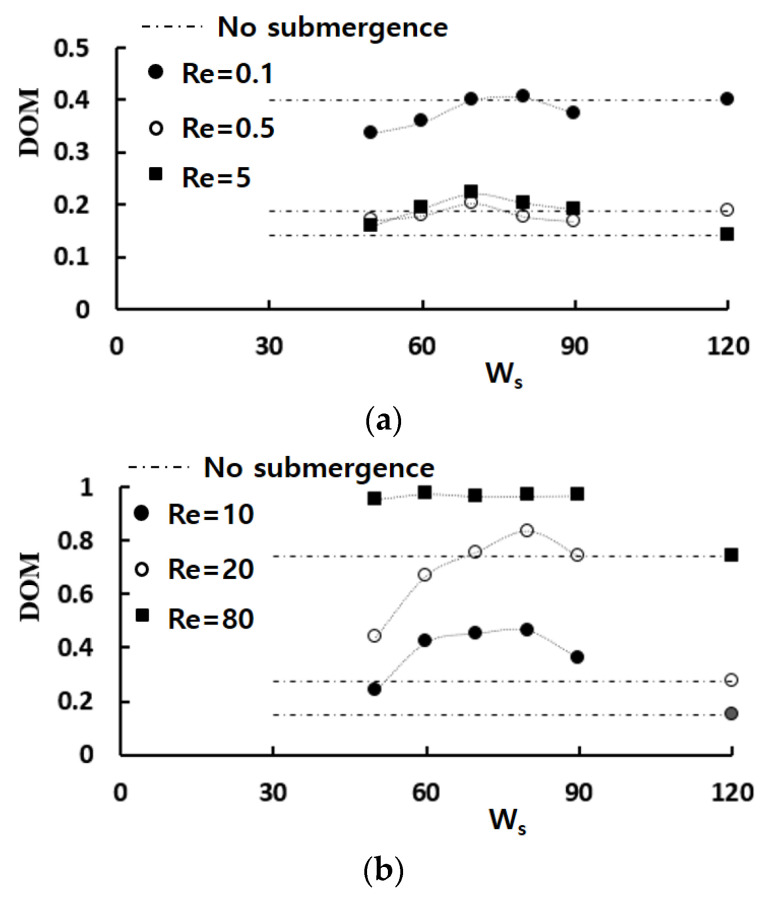
Effects of the height of submerged structure on the DOM: (**a**) Molecular dominance and transition regime of mixing and (**b**) Convection dominance regime of mixing.

**Figure 8 micromachines-13-01050-f008:**
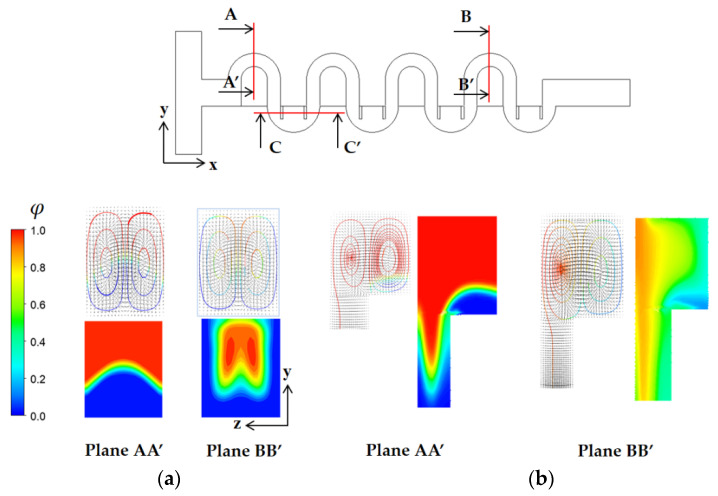
Comparison of projected streamlines and concentration distribution for Re = 10: (**a**) No submergence and (**b**) Submerged, *W_s_* = 60 μm.

**Figure 9 micromachines-13-01050-f009:**
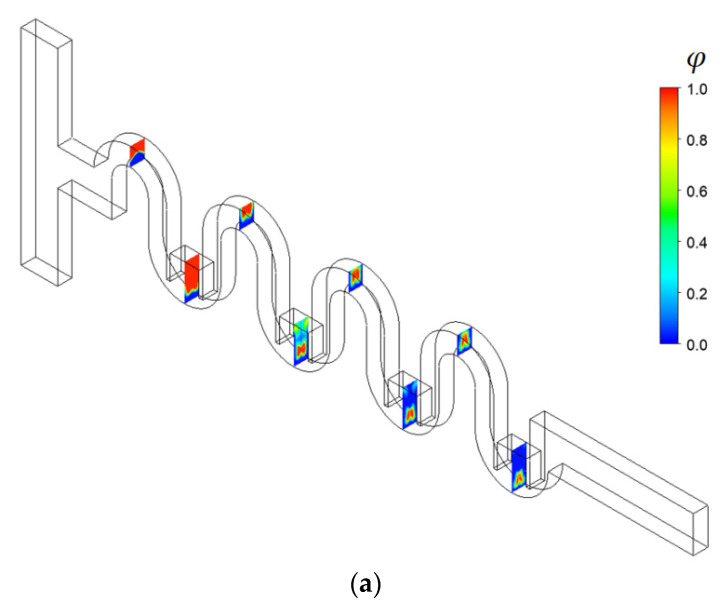
Comparison of mixing evolution along the micromixer for Re = 10: (**a**) No submergence, (**b**) Submerged, *W_s_ =* 60 μm.

**Figure 10 micromachines-13-01050-f010:**
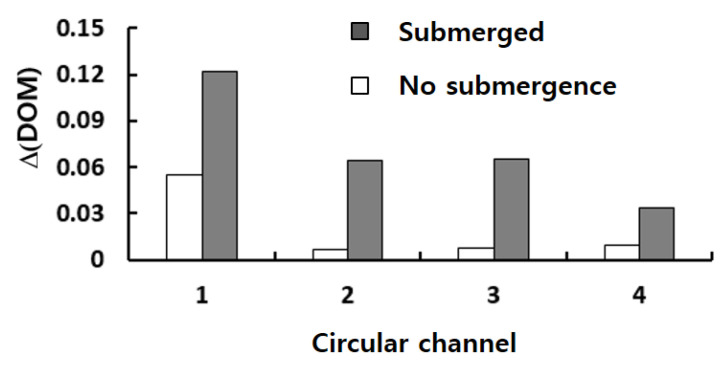
Comparison of the DOM increment in each circular channel for Re = 10.

**Figure 11 micromachines-13-01050-f011:**
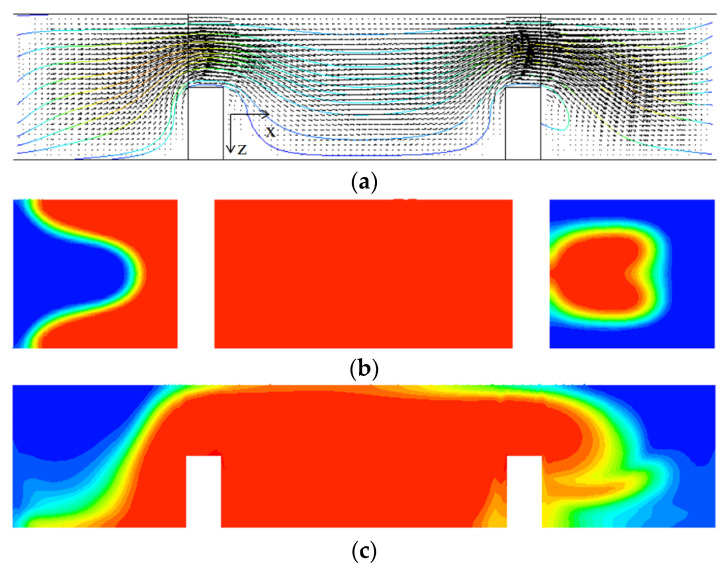
Projected streamlines and velocity vector on the three cross sections for Re = 10 structures: (**a**) Projected streamlines for the submerged case, *W_s_ =* 60 μm, (**b**) Concentration distribution for no submergence case, and (**c**) Concentration distribution for the submerged case, *W_s_ =* 60 μm.

**Figure 12 micromachines-13-01050-f012:**
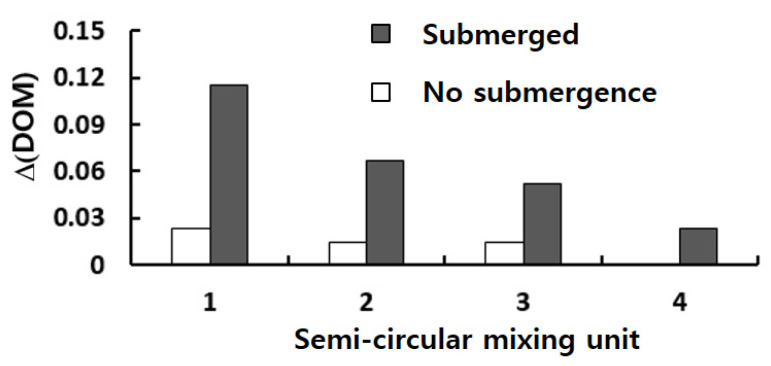
Comparison of the DOM increment in each semi-circular mixing unit for Re = 10.

**Figure 13 micromachines-13-01050-f013:**
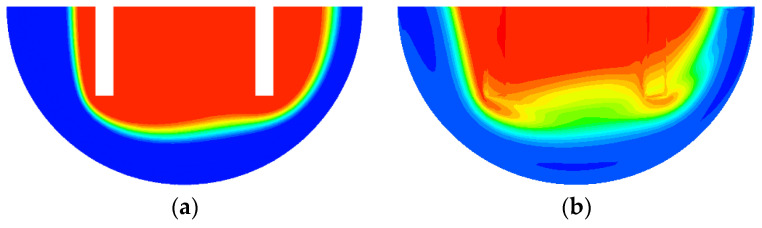
Comparison of concentration distribution in the first semi-circular mixing unit for Re = 10: (**a**) No submergence and (**b**) Submerged, *W_s_* = 60 μm.

**Figure 14 micromachines-13-01050-f014:**
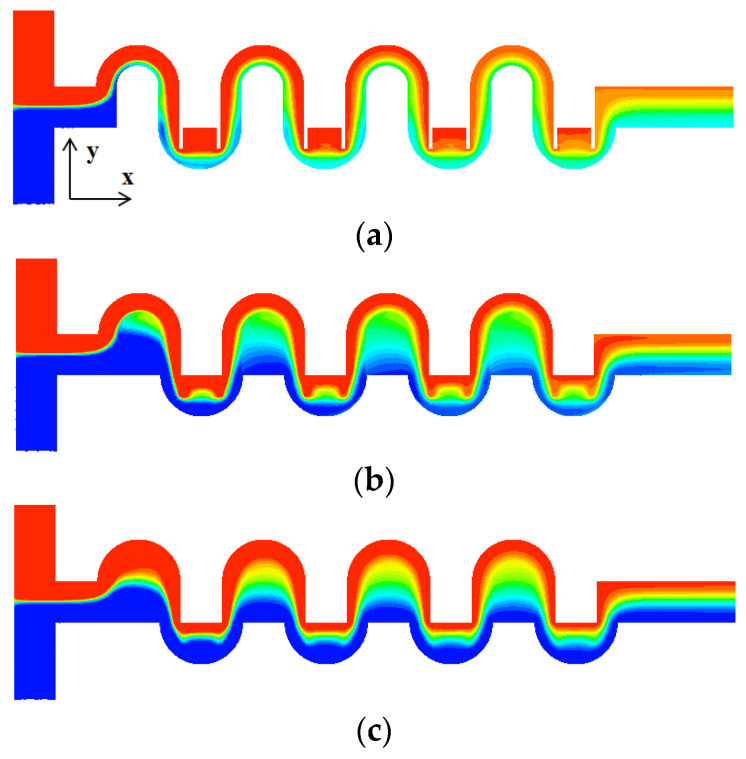
Concentration distribution on three planes for *W_s_* = 60 μm and Re = 0.1: (**a**) At z = 30 μm, (**b**) At z = 60 μm, and (**c**) At z = 90 μm.

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
