# Peer review of "Mixing Enhancement of a Passive Micromixer with Submerged Structures"

_micromachines, 2022, doi:10.3390/mi13071050_

Round 1
Reviewer 1 Report
The manuscript by Juraeva and Kang describes a passive micro-mixer submerged structures that was simulated numerically over a range of Reynolds numbers. Although the technique described and results produced are very interesting, the report as it currently stands is not of sufficient quality to warrant publication due to the reasons mentioned below and it requires a major revision before acceptance. Below are some of the comments I have on what is presented in the manuscript:
The information in the abstract should be reorganized to make it easier for the reader to follow. The following sentence “was modified to submerge some structures,” in lines 8-9 isn’t very clear on its own and the sentence “The two submerged structures are the Norman window and rectangular baffles” in line 11 should come directly after it.
In the abstract, “degree of mixing” should precede the first instance of “DOM”.
The manuscript should be revised to improve the grammar throughout the manuscript. There are some grammatical mistakes in the manuscript that should be avoided (e.g. “With” shouldn’t be capitalized in the title and the title shouldn’t have an endpoint, “and was simulated numerically its mixing performance” line 9 ,“micro-size of devices” line 30, “split and recombine of streams” line 52, “are ease to obtain” line 83, etc.)
Authors need to clarify what range of Reynolds number do they mean in the following statement and how different is that from the range they are using in this manuscript “However, these flow characteristics are dominant only in a limited range of the Reynolds number, and/or requires higher pressure load.” In lines 62-63.
Are there references to support the following claim “In general, the submergence of planar structure requires less pressure load, compared with that of no submergence planar design.” or was this observed by the authors in their previous work results.
The reference [26] in line 91 seems to be incorrectly placed. Also, Tsai et al. are referenced as [32] in the manuscript; however, Figure 4 cites them as [25]. The authors are required to revise their reference list to make sure that references are appropriately cited in text.
The authors have recently published papers [14, 23, 24] on the same topic with a similar micro-mixer structure, they are required to show how this paper builds on their previous results and not just extend the results of [23,24]. The introduction section should include a clearer purpose for this paper. What is still missing in the field and how does this paper bridge the gap?
The authors need to provide reasoning in the text on:
i. Why is the Reynolds number range from 0.1 to 80 is important to investigate?
ii. Is the volumetric flow rate from 1.3 μL/min to 1012.8 μL/min practical in the field of microfluidics even though there’s almost a 3 order of magnitude difference?
iii. Why were the two submerged structures chosen to be the “Norman window” and “rectangular baffles” and not other structures.
In line 172, the baffle width is mentioned as 45 μm; however, it is shown as 40 μm in Figure 3. The authors are required to correct this.
The statement in line 180-181 “About 1 million of the computational cells is large enough to obtain a numerical solution with a reasonable accuracy.” is misleading. 1 million cells may be enough for this particular case but should not be generalized. The authors are encouraged to clarify this point in the text.
In line 202, the authors claim that “The discrepancy between the experimental data and numerical solution is less than 4%” which they refer to as “acceptable” in line 201. What’s the maximum value of discrepancy allowed to still be considered acceptable? And is this maximum value based on common accepted practices or the authors’ own engineering judgments.
Higher resolution figures should be uploaded. Some text in figures is pixelated.
The mesh size value is missing in “The corresponding number of mesh varies from to .” in lines 225 and 226.
In line 246, the authors claim that “A new passive micro-mixer with submerged structures was proposed”; however, the structure seems very similar to the one they have previously shown in [23]. The authors are encouraged to clarify how the micro-mixer in this study “significantly” differs from the ones they have previously developed to be considered “new”.
The sentence in lines 307-308 “On the contrary, one vortex bursts into the submerged space for the submerged case.” is not very clear. Authors are encouraged to rewrite this sentence to make their point clearer.
In line 314, the authors are encouraged to check if they meant “Figure 9” and not “Figure 10”.
The value of “Ws” should be added to Figure 15. Was it also taken as Ws = 60 μm. An x-y coordinate symbol may also be added to this figure to make it easier to understand similar to Figure 1a.
In line 431, “Reynolds numbers, Re = 50.1 ~ 80.” should be “Re = 0.1 ~ 80.”
The authors are encouraged to add the third mixing regime aside from “molecular diffusion” and “convective mixing” in line 432.
Finally, I would like to state that although the approach and results are very interesting to the readers of Micromachines, the manuscript submitted has many subjective statements and requires a major revision. I recommend the authors to revise the manuscript to provide a clearer purpose for this paper in the introduction, show how their approach builds on their previous work in [23, 24] and is different from it, provide a clearer comparison between the submerged and unsubmerged structures, provide a better physical understanding and explanation on why this enhancement in results is being achieved compared to the results in the unsubmerged micromixer, and how their results can be further used.
Author Response
Thanks for reviewing the paper.
Please refer to the attached file.

Reviewer 2 Report
Comment 1: Abstract should be revised. For example, the Re range mentioned twice at the abstract without any reason. Also, DOM, should be written as degree of mixing (DOM) and then authors could use then only the term DOM.
Comment 2: For line range 47-58 (and several other references at different positions) , authors use Santana et al. [17], Sotowa et al. [18]. They should replace it with “Among them Santana claim that………[17]” or as already use in line 84.
Comment 3: Authors sometimes use the term Figure or Fig (lines 97 and 99). Must follow the instructions for authors.
Comment 4: At line 174 the viscosity should be in S.I. and written as the diffusion coefficient in the same line for example. Also, at line 248.
Comment 5: At line 226 something is missing the sentence is not completed.
Comment 6: Suggestion to authors is to remove Fig.6 since the above paragraph is clear and there is no need for the existence of Fig.6. This is only a suggestion.
Comment 7: Authors used a range of inlet velocities. The velocities for both inlets were equal for each simulation as mentioned in line 248. It is recommended to include some latest related research to enhance their results. Since for velocity ratio equal to 1 succeed high DOM. Hence for higher velocity ratio should be expect higher DOM. Some references are listed below:
https://www.mdpi.com/2673-4931/2/1/65
https://iwaponline.com/aqua/article/doi/10.2166/aqua.2022.080/88806/Mixing-of-Fe3O4-nanoparticles-under
Comment 8: At references section some of the references haven’t the same space between them.
Comment 9: Figure 7(b) should be discussed further. ΔP is proportional to Re?
Author Response

(The authors gave the same response as above.)

Round 2
Reviewer 1 Report
The authors have thoroughly improved the manuscript according to initial comments and recommendations provided earlier. However, the authors are encouraged to address the following very minor errors:
The new paragraph added in line 110 “Many micromixers… 1012.8 μL/min.” should be moved upwards and placed before the paragraph starting at line 80.
The word “decreases” should be “increases” in line 230 “becomes smaller as the Reynolds number decreases”
The word “new” should be removed in line 267 “A new passive micromixer with submerged structures was proposed,” as the authors have previously shown this micromixer in their previous work [26].
There are still some clear grammatical mistakes and typos throughout the manuscript that should be avoided.

Reviewer 2 Report
The authors have addresed all the points raised by the reviewers and made the appropriate changes. Thus I recommend the paper for publication